# Discovery of colossal Seebeck effect in metallic Cu$_2$Se

Dogyun Byeon[1], Robert Sobota[1], Kévin Delime-Codrin[1], Seongho Choi[1], Keisuke Hirata[1], Masahiro Adachi[2], Makoto Kiyama[2], Takashi Matsuura[2], Yoshiyuki Yamamoto[2], Masaharu Matsunami[1] & Tsunehiro Takeuchi[1]

Both electrical conductivity $\sigma$ and Seebeck coefficient $S$ are functions of carrier concentration being correlated with each other, and the value of power factor $S^2\sigma$ is generally limited to less than 0.01 W m$^{-1}$ K$^{-2}$. Here we report that, under the temperature gradient applied simultaneously to both parallel and perpendicular directions of measurement, a metallic copper selenide, Cu$_2$Se, shows two sign reversals and colossal values of $S$ exceeding ±2 mV K$^{-1}$ in a narrow temperature range, 340 K < $T$ < 400 K, where a structure phase transition takes place. The metallic behavior of $\sigma$ possessing larger magnitude exceeding 600 S cm$^{-1}$ leads to a colossal value of $S^2\sigma = 2.3$ W m$^{-1}$ K$^{-2}$. The small thermal conductivity less than 2 W m$^{-1}$ K$^{-1}$ results in a huge dimensionless figure of merit exceeding 400. This unusual behavior is brought about by the self-tuning carrier concentration effect in the low-temperature phase assisted by the high-temperature phase.

[1] Toyota Technological Institute, Hisakata 2-12-1, Tempaku, Nagoya 468-8511, Japan. [2] Sumitomo Electric Industries, Ltd., Konyo Kita 1-1-1, Itami, Hyogo 664-0016, Japan. Correspondence and requests for materials should be addressed to T.T. (email: t_takeuchi@toyota-ti.ac.jp)

The thermoelectric (TE) materials have been systematically and comprehensively studied during the past several decades, primarily due to their capability of converting waste heat into useful electrical power[1,2]. The efficiency of TE energy conversion is an increasing function of the dimensionless figure of merit, $ZT = S^2\sigma T / \kappa$, where $S$, $\sigma$, $T$, and $\kappa$ stand for the Seebeck coefficient, electrical conductivity, absolute temperature, and thermal conductivity, respectively. Up to now, numerous TE materials have been discovered, some of them showing ZT higher than unity[3–7], though, the overall performance and efficiency are not sufficient, and thus, cannot be widely used for a variety of applications.

The largest value of ZT ever reported is ZT = 2.6 discovered in a single crystal SnSe[3]. Unfortunately, the temperature of the largest ZT was 900 K, where competitive methods used for power generation possess much higher efficiency of energy conversion. In the middle-low temperature range of 300~400 K, where a large amount of waste heat is emitted into the ambient environment, we do not have any other methods to effectively generate electrical power, and therefore, the wide use of the mentioned thermoelectric materials is generally expected. Up to now, $Bi_2Te_3$-based materials hold the best performance in this temperature range with a ZT of up to 2.0[8]. However, at the same time, it has to be strongly emphasized that these ZT values are still controversial[9]. Nearly the similar value of ZT was recently reported for $Cu_2Se$ at about 400 K[7].

All these materials are characterized by the very small lattice thermal conductivity less than $1.0\,W\,m^{-1}\,K^{-1}$ together with a relatively large power factor $PF = S^2\sigma$ in close relation with electronic structures suitable for thermoelectric materials and optimal carrier concentration. Both $S$ and $\sigma$ are functions of carrier concentration but possess an opposite behavior: the former increases with a decreasing number of carrier concentration whereas the other decreases. Therefore, the optimal carrier concentration was investigated for many materials, and realized that a few hundreds of $\mu V\,K^{-1}$ in Seebeck coefficient together with a few of $m\Omega\,cm$ in electrical resistivity, and a few of $mW\,m^{-1}\,K^{-2}$ in power factor, would be the best suited for thermoelectric materials.

In this work, we report a new surprising discovery of colossal values of Seebeck coefficient in "metallic" $Cu_2Se$ in a temperature range of 340−400 K, where an order−disorder structure transition takes place, possessing unusually high values of power factor exceeding $2.3\,W\,m^{-1}\,K^{-2}$. These values were observed under the unusual temperature gradient applied not only to the direction parallel to the Seebeck measurement but also perpendicularly at the same time. Supported by the very small lattice thermal conductivity of less than $1.8\,W\,m^{-1}\,K^{-1}$, the estimated ZT value exceeds 450, despite that the present temperature gradient is not applicable to obtain the ordinary used relation between ZT and $\eta$, which represents the efficiency of energy conversion for $\pi$-type thermoelectric power generators. These colossal values of $S$, PF, and ZT were attained in the low-temperature phase as a result of the self-tuning carrier concentration caused by the influence of the simultaneously persisting high-temperature phase.

## Results

**Wide range of temperature required for the phase transition**. Structure analysis (Fig. 1a, b) performed on the basis of synchrotron radiation diffraction measurements proved that $Cu_2Se$ at room temperature unambiguously corresponds to the hexagonal structure (space group: $R\overline{3}m$, Pearson Symbol: $hR6$) of the low-temperature α-phase[10]. With increasing temperature, the hexagonal structure continuously undergoes a phase transition to a cubic high-temperature β-phase ($Fm\overline{3}m$, Pearson symbol: $cF12$)

over a high-temperature interval from 320 to 390 K[11]. Both the crystal structures are shown in Fig. 1c. The volume fractions of two phases gradually vary with an increasing temperature as shown in Fig. 1d. A characteristic temperature of the phase transition, where the high-temperature phase overcomes the other, appears at the highest temperature of the phase transition around 390 K. This is tightly connected with the drastic variation in electrical resistivity and thermal conductivity which will be mentioned later. The calculated XRD patterns using the Rietveld analysis at room temperature as well as 473 K are shown in Fig. 1a, b with lattice constants and $R_{wp}/R_p$-factors. More detailed information about samples and Rietveld analysis are shown in Supplementary Figure 14, Supplementary Table 2, and Supplementary Note 8.

**Unusual behavior of the Seebeck coefficient**. The most important and fascinating discovery of our work is an extremely large magnitude of Seebeck coefficient, $S(T)$, found in the metallic $Cu_2Se$ in the experimental setup shown in Fig. 2a. The typical example of data measured by our experimental setup together with reference data[7] are displayed in Fig. 2b, c. Additional information about the Seebeck measurement is described in Supplementary Figures 1−3, 5−10, and Supplementary Notes 1, 2, 4−6. The positive values of the Seebeck coefficient measured at low temperatures below 330 K drastically decreased to be negative with an increasing temperature. After becoming minimal of $-4347\,\mu V\,K^{-1}$ at about 347 K, another sudden sign reversal occurred creating a positive peak of $1982\,\mu V\,K^{-1}$ at 349 K. Further increase of a temperature led to a plateau of ~$220\,\mu V\,K^{-1}$ over a relatively wide temperature range up to ~394 K. Afterwards, $S(T)$ values were reduced to ~$90\,\mu V\,K^{-1}$ with a small positive temperature coefficient. The sudden change in $S(T)$ would be related to an unusual temperature dependence of chemical potential near the conduction and valence bands as discussed later.

This very unusual behavior of the Seebeck coefficient was observed for not only the sample shown in this paper but also many other samples with slightly different compositions or even a small amount of partial element substitutions. Besides, we observe almost the same behavior in Seebeck coefficient of many different samples, and realized that the peak magnitude of Seebeck coefficient sensitively varies with a sample thickness and composition. Typical examples of slightly smaller magnitude of Seebeck coefficient with the similar temperature dependence as measured on several $Cu_2Se$ samples having almost the same chemical composition and thicknesses are shown in Fig. 3a–e. As it can be seen that although the magnitude of the negative peak is observed statistically to be about $-1200\,\mu V\,K^{-1}$ with a small aberration in the peak temperature at the phase transition, the ultimate Seebeck coefficient data as presented in Fig. 2b was three times larger than usually observed by our setup. This means that the behavior is evidently appearing at the point of the phase transition and gives reproducible outcome excluding possible artifact occurrences caused by the setup, measurement protocol and/or analysis method. The data reproducibility is also explained in Supplementary Figures 4−6, and Supplementary Note 3.

The p-n-p-type sign reversals with colossal magnitude of Seebeck coefficient were also reported for AgCuS at 350−450 K[12], $AgBiS_2$ nanocrystals at ~560 K[13], and $Ag_{10}Te_4Br_3$ at ~380 K[14]. However, such behaviors were measured under an ordinary temperature gradient. In these cases, phase transitions between a low-temperature phase to an intermediate phase and the intermediate phase to a high-temperature phase were considered as the origin of p-n and n-p transition, respectively[12]. In sharp contrast to the p-n-p transitions reported for these materials, the

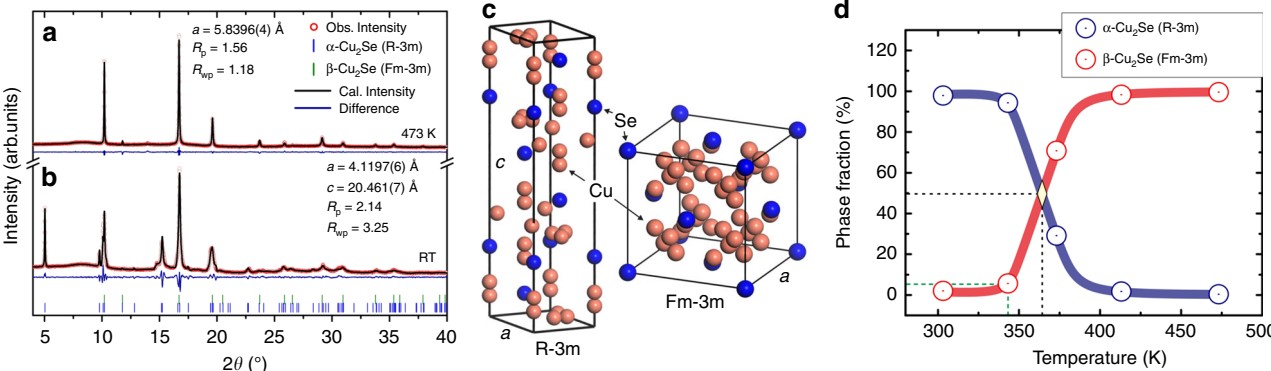

**Fig. 1** Structure analysis on $Cu_2Se$. **a** Synchrotron radiation powder diffraction patterns at room temperature (RT) and **b** 473 K together with Rietveld analysis data including space groups $R$-factors and lattice constants. **c** High/low-temperature crystal structure of $Cu_2Se$. **d** Phase fractions of $\alpha$-$Cu_2Se$ and $\beta$-$Cu_2Se$ as obtained from Rietveld analysis

$Cu_2Se$ samples presented in this study do not show any evidence of an intermediate phase. Besides, our samples do not possess p-n-p transformation under circumstances of the ordinary temperature gradient applied along the direction of measurement but only under the unusual temperature gradient both parallel and perpendicular to the Seebeck measurement direction as explained in Supplementary Figures 1−3, and Supplementary Notes 1, 2.

It would be also worthwhile to mention that the unusual temperature dependence of the Seebeck coefficient observed in this study is very similar to the carrier concentration dependence of the Seebeck coefficient of semiconducting materials in the point that it possesses large negative and positive peaks with a sudden sign reversal in a very narrow range.

**Colossal values of power factor and figure of merit.** Electrical resistivity, $\rho(T)$, measured with the same experimental setup as the Seebeck measurement, plotted in Fig. 4a, shows typical metallic behavior over a wide temperature range, except for an unusual increase in a temperature range of 300–340 K (see also Supplementary Figures 4, 6, and Supplementary Note 3). The $\rho(T)$ begins smoothly increasing from 300 K and drops rapidly after becoming maximal at ~343 K. The rather small value of $\rho(T)$ together with the large magnitude of $S$, as obtained in Fig. 2b, naturally leads to surprisingly large values of power factor at the peaks of $S(T)$: 0.18~2.3 W $m^{-1}$ $K^{-2}$ and 0.06~0.5 W $m^{-1}$ $K^{-2}$ for n-type and p-type, respectively (Fig. 4b). These values are definitely much larger than a few mW $m^{-1}K^{-2}$ of typical thermoelectric materials.

The observed tendency of $\rho(T)$ is almost the same as the previously reported results, except for the slightly lower temperature of the peak[7,15,16]. We considered that the difference in the peak temperature in electrical resistivity would be related to the vertical temperature gradient of the sample in the Seebeck measurement setup. To confirm this consideration, we additionally measured $\rho(T)$ in a vacuum tube furnace without any temperature gradient applied, realizing that the $\rho(T)$ peak temperature was increased to 379 K, showing rather good consistency with the reference data. A small aberration of the peak temperature and the peak broadening in our data are plausibly caused by a minor composition difference as it was clearly shown in the previously reported paper[17] that $\rho(T)$ strongly depends on the Cu deficiency. Noting also that although the nominal composition of the presented sample type is identical to the reference in Fig. 4a, the real composition can vary due to the different preparation method and the high Se volatility, and thus, the real composition after the synthesis process is altered.

Notably, the characteristic temperature of XRD also coincides with the temperature of $\rho(T)$ reduction.

Thermal conductivity, $\kappa(T)$, specific heat, $C_p(T)$, and thermal diffusivity, $D(T)$, measured by the laser flash method was shown in Fig. 4c, d. $D(T)$ and $C_p(T)$ displayed an opposite behavior at the phase transition. The total $\kappa(T)$ maintained increasing near the phase transition since the jump in $C_p(T)$ was much higher than the drop of $D(T)$. The very unusual increase of $\kappa(T)$ at about 396 K was reported also by Kim et al.[18], though the mechanism causing this behavior has not been revealed yet. As can be obviously seen, the difference between our $\kappa(T)$ and the reference data[18] is again caused by the composition difference as in the case of $\rho(T)$, being highly sensitive to the deficiency of Cu. The maximum value of $\kappa(T)$ was ~3 W $m^{-1}$ $K^{-1}$ near the phase transition, whereas the smallest value attained was ~1.7 W$m^{-1}K^{-1}$, resulted from the extremely low lattice thermal conductivity of ~1.0 W $m^{-1}$ $K^{-1}$ $(= \kappa(T) - (L_0\sigma T - S^2\sigma T)$, where $L_0 = \pi^3 k_B^2/(3e^2)$ represents a constant known as the Lorenz number. Note here that the second term $S^2\sigma T$ in the electron thermal conductivity[19–21] is negligibly small to be generally ignored, but has to be taken into account for the materials possessing a large power factor PF = $S^2\sigma$ such as the material presented in this study. It is strongly believed that this very small lattice thermal conductivity would be realized by the anharmonic oscillations of the lattice in association with the split sites of copper[6,18].

The temperature range of the sharp peak in $\kappa(T)$ and $\rho(T)$ at around 396 K is slightly higher than 344~346 K, where the extraordinary Seebeck coefficient was detected. This difference between the two temperature ranges allowed us to obtain exceptionally large values of ZT (as shown in Fig. 4e−g). The values of maximum ZT reaching 471 were obtained in association with the huge magnitude of the negative Seebeck coefficient −4347 μV $K^{-1}$, rather small electrical resistivity 0.8 mΩ cm, and small thermal conductivity 1.7 W $m^{-1}$ $K^{-1}$. This colossal value of ZT is nearly 170 times superior to the highest values ever reported for SnSe[3]. Even when the magnitude of Seebeck coefficient is reduced to −1200 μV $K^{-1}$ at the peak, the maximum ZT value still exceeds 30. The value of ZT with the positive sign of the Seebeck coefficient also reached 10~100, indicating that a thermoelectric device with a pair of p-type and n-type materials could be produced using two $Cu_{2-\delta}Se$ samples possessing slightly different phase transition temperatures. Notably, the plateau of the Seebeck coefficient also resulted in a high magnitude of ZT ~1.2, being kept over the relatively wide temperature range of 350 K < T < 390 K.

It should be also emphasized that a relatively large value of ZT ~2 was reported previously for $Cu_2Se$ using the value of

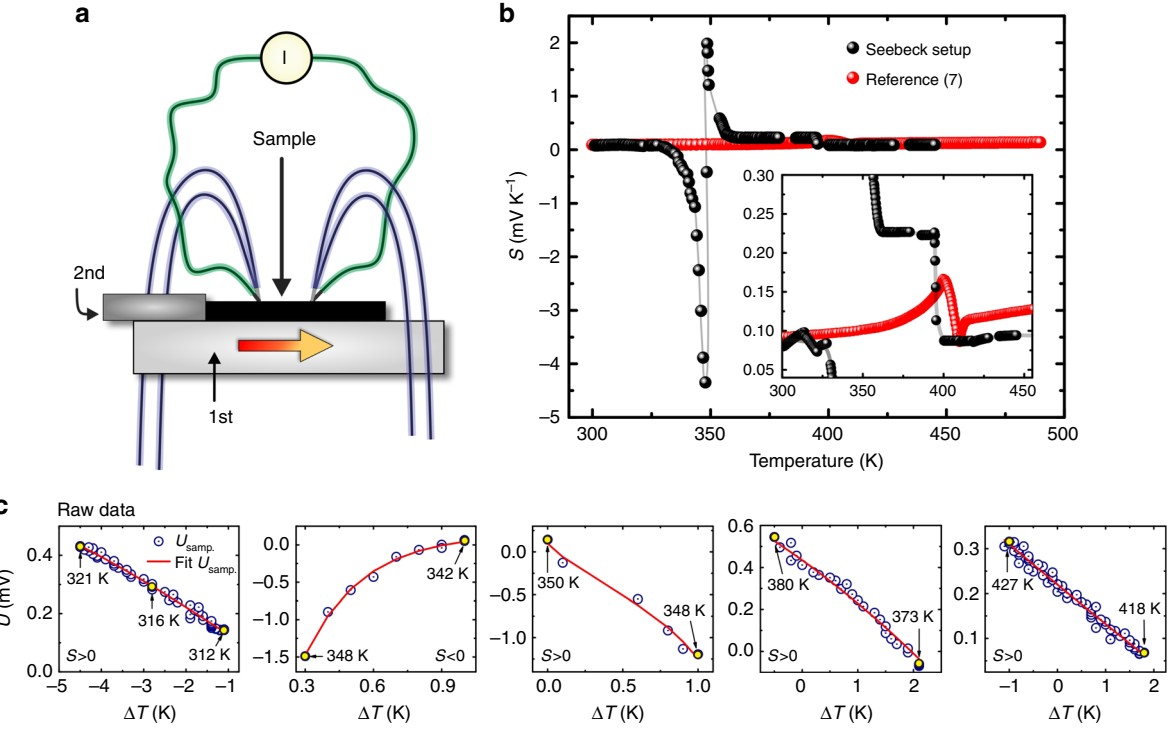

**Fig. 2** Seebeck coefficient of $Cu_2Se$. **a** Schematic drawing of the Seebeck experimental setup. **b** Seebeck coefficient results with its zoom in the inset, and reference data[7]. **c** Raw data of the thermal electromotive force as obtained by the setup

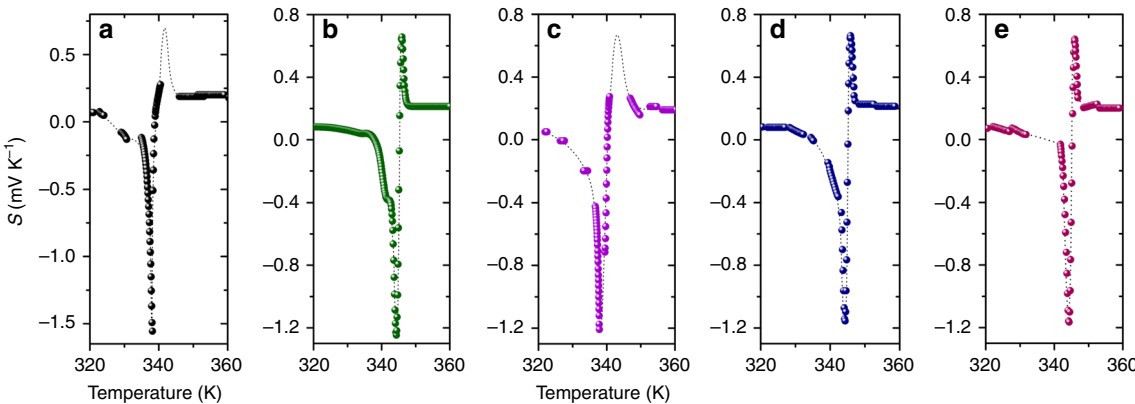

**Fig. 3** Reproducibility of Seebeck coefficient data in $Cu_2Se$. **a**−**e** Representative series of Seebeck coefficient data acquired on $Cu_2Se$, prepared by the same method and having the almost identical chemical composition and thickness of 1.3 mm, in a range of the abnormal behavior

$C_p(T)$ obtained from the Dulong-Petit law[7]. If we use the same method to calculate ZT, the values increase up to ZT ~1000 and 400 for n-type and p-type behavior, respectively. We also have to consider the peak shift of thermal conductivity in the Seebeck measurement setup. If the peak in $\kappa(T)$ coincided with that of $S(T)$, the value of ZT could become 1.76 times smaller. Even in such a case, ZT is still kept extremely large at 260 and 56 for n-type and p-type behavior, respectively.

**Mechanism for the colossal Seebeck effect with two sign reversals**. Despite the intensive research on $S(T)$ of $Cu_2Se$ by many different groups[6,7,15,17], none of them has published the unusual behavior presented in this paper. We realized that the key and essential point is hidden in our experimental setup designed to apply a horizontal temperature gradient, though, a vertical temperature gradient due to the experimental setup geometry is

also produced simultaneously. The temperature difference between two measuring points was kept at 0~5 K for precisely determining the Seebeck coefficient, while the temperature difference between the top and heated bottom surfaces was measured to be ~40 K. This indicates that the high-temperature phase appeared in the sample bottom while the low-temperature phase remained in the top surface in the temperature range of the unusual Seebeck coefficient behavior. Besides, in the middle of the sample, two phases coexist together as a mixed phase.

The difference of $\rho(T)$ between two different sample setups is qualitatively accounted for with the scenario of two phases coexisting in the sample. The extremely large Seebeck coefficient suggests that the low-temperature phase in the top surface of the sample supposed to possess low carrier concentration leading to the higher $\rho(T)$ than the mixed phase in the middle and the high-temperature phase in the bottom. If that was the case, the

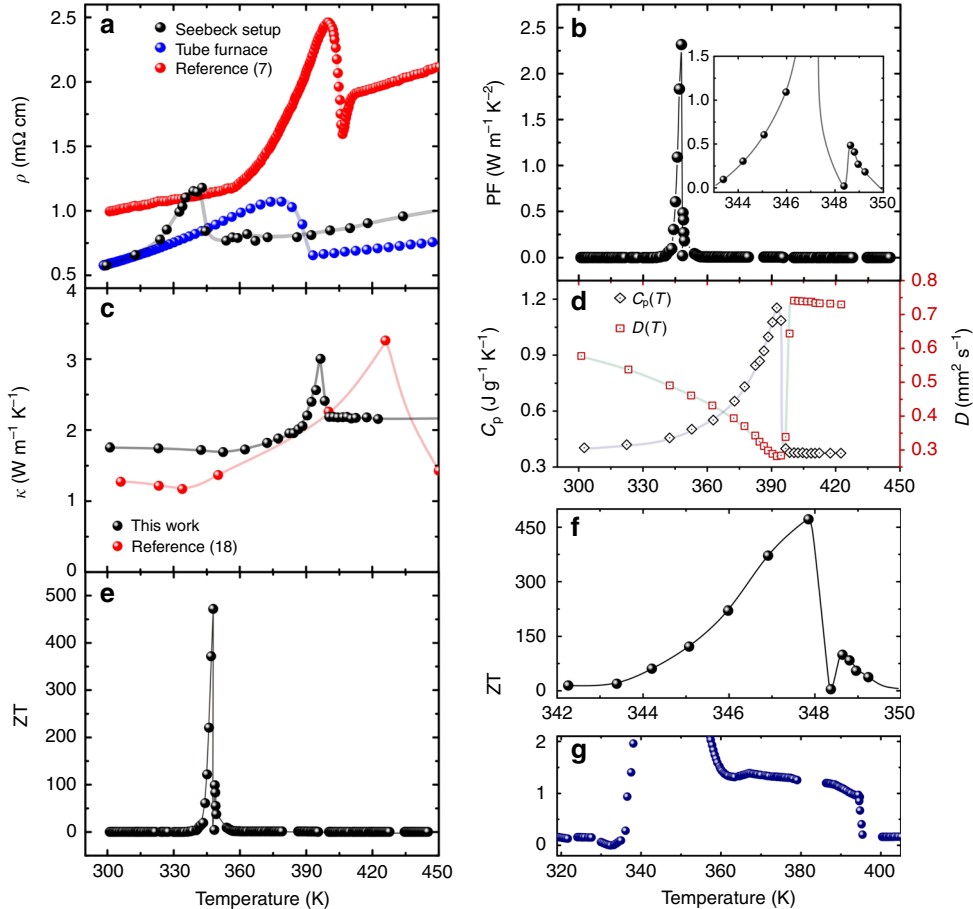

**Fig. 4** Temperature dependences of thermoelectric properties of $Cu_2Se$. **a** Electrical resistivity, $\rho$, together with the reference data of the nominal composition $Cu_2Se$ [7]. **b** Power factor, PF, displaying two peaks related to negative and positive $S(T)$. **c** Thermal conductivity, $\kappa$, and reference data [18]. **d** Specific heat, $C_p$, together with the thermal diffusivity, $D$. **e** Dimensionless figure of merit ZT. **f** Region of abnormal behavior in ZT temperature dependence showing two peaks. **g** The plateau observed in ZT temperature dependence

electrical current did not flow completely inside of the high resistive low-temperature phase but instead, the main fraction passed through the less resistive part situated in the middle and bottom parts of the sample. This effect made the peak of $\rho(T)$ at the lower temperature than that observed in the standard vacuum furnace where homogeneous temperature distribution was present. We also strongly believe that the thickness of the low-temperature phase in the top surface was very thin because the magnitude of $\rho(T)$ in the Seebeck setup was only slightly higher than the values measured in the standard tube furnace creating a homogeneous temperature distribution in the samples.

For interpreting the unusual Seebeck coefficient, we assumed that the chemical potentials of copper ions and conduction electrons in the low-temperature phase could be different from those in the high-temperature phase. In such a case, copper ions and electrons slightly move from one of the phases to the other so as to reach the energy equilibrium between both phases. The effect of the chemical potentials should change the carrier concentration of each phase. With an increasing temperature, the sample bottom starts transforming to the high-temperature phase leading to an increase of the electron concentration in the low-temperature phase situated above due to the difference between the chemical potentials of electrons and/or copper ions. The number of electrons in the low-temperature phase in the top surface is further enlarged with the increased volume fraction of the high-temperature phase. Under the effect of the chemical potentials on the carrier concentration, the positive Seebeck

coefficient of the low-temperature phase below 320 K becomes negative at around 330 K and reaches $-4347\ \mu V\ K^{-1}$ at 347 K. In the higher temperature range above 347 K, the chemical potential of either electrons or copper ions leads to a reduction of the electron concentration of the low-temperature phase, and the positive peak of the Seebeck coefficient is created. Besides, one of the factors would be sustained in the temperature range of $350\ K < T < 390\ K$ forming the plateau of the Seebeck coefficient. Afterwards, the whole sample becomes high-temperature phase exhibiting the same tendency of the Seebeck coefficient as in the refs. [7,15,17,22,23].

For confirming the validity of this scenario, we calculated the electronic structure of the low-temperature phase by means of first-principles band calculations using the distributed package programs, VASP [24,25] and WIEN2K [26]. The pseudo-potential method, used in VASP allowing us to reduce the calculation time, was used to optimize the crystal structure model involving the combined copper sites free from the splitting. After the structure optimization, the full-potential linearized augmented plane wave method within Wien2K was employed to obtain detailed information on the electronic structure (Fig. 5a, b, Supplementary Figure 15, Supplementary Table 3, and Supplementary Note 9). Subsequently, the chemical potential dependence of the Seebeck coefficient at 345 K, in the temperature vicinity of which the positive and negative peaks in the Seebeck coefficient data were observed, was calculated using the Boltzmann transport equation with the BoltzTrap code [27] as

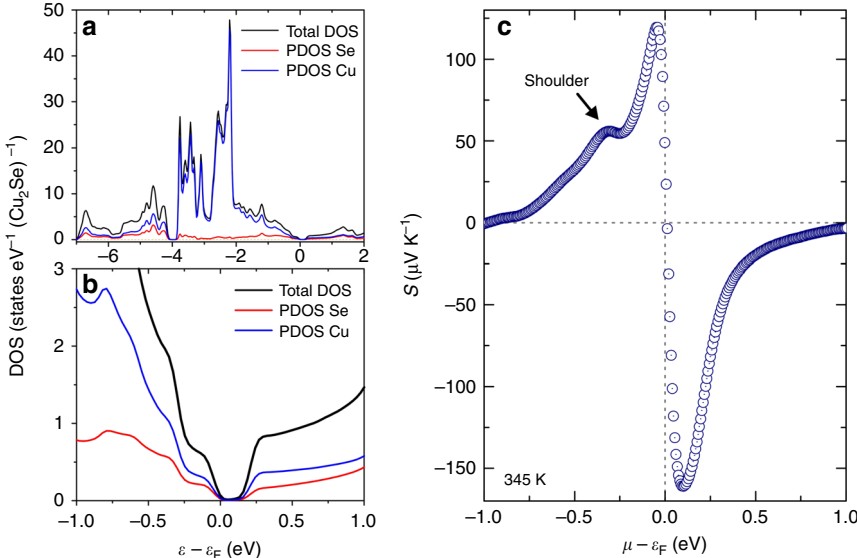

**Fig. 5** Calculated electronic density of states and Seebeck coefficient of the simplified $Cu_2Se$. **a** Electronic density of states calculated by Wien2k. **b** Zoom-in of the near gap region. **c** The calculated Seebeck coefficient as a function of chemical potential at 345 K

shown in Fig. 5c. Notably, despite that the hypothetic structure model reduced the calculated gap size to make the magnitude of the Seebeck coefficient at the peaks definitely smaller, the calculated Seebeck coefficient as a function of chemical potential, $S(\mu)$, shows fairly good consistency with the experimental outcome $S(T)$ in the points that (a) negative peak possessed a larger magnitude than the positive peak and (b) a shoulder is observable at the low energy side of the positive peak in the same manner as we observed the positive plateau of Seebeck coefficient in $350\,K < T < 380\,K$. These similarities certainly lend a great support to our scenario: self-tuning carrier concentration under the influence of two-phase coexistence.

One may realize that the plateau in $S(T)$ is certainly wider than the shoulder in $S(\mu)$. This difference can be explained in such a way that the electronic density of states is relatively large in the energy range where the shoulder in $S(\mu)$ is obtained. The large electronic density of states prevents the drastic variation of chemical potential, even though the carrier concentration varies significantly with the temperature. This mechanism effectively widens the plateau in $S(T)$.

It is naturally expected that this unusual behavior of the Seebeck coefficient could be universally observed in semiconducting materials possessing a structure transition, provided that the experimental geometry used in this study is employed. The group of (Ag, Cu)$_2$(S, Se, Te), that includes the sample type as presented in this study, is one of the typical examples. We consider that the intensive research on these materials could lead to further unusual behaviors of electron and thermal transport properties.

Before ending, we should make comments on the potential of $Cu_{2-\delta}Se$ for practical applications. The equation leading to the efficiency of energy conversion in a thermoelectric generator, $\eta$, that increases with an increasing ZT, was derived on the basis of a model involving π-type junctions made of two thermoelectric materials. In this scenario, a temperature gradient is applied to the thermoelectric materials simply along the direction of electrical current. The temperature distribution of the samples, in the setup introduced in this study, was certainly different from the case of the π-type module. Therefore, ZT is no longer valid to estimate $\eta$ in our case. Nevertheless, we should emphasize that the large $S(T)$, small $\rho(T)$ and consequently obtained PF, as measured using the same electrodes, can be certainly applicable

for an effective energy conversion between heat and electricity, and nearly the same value of $\eta$ estimated from ZT should be obtained with an appropriate temperature distribution.

## Discussion

In this study, we discovered that the large values of PF and ZT exceeding $2.3\,W\,m^{-1}\,K^{-2}$ and 470, respectively, are observable in metallic $Cu_2Se$ at around 350 K together with the unusual temperature dependence of Seebeck coefficient. The DFT band calculations revealed that this colossal Seebeck effect is brought about by the self-tuning carrier concentration effect in association with the two-phase coexistence during the phase transition.

## Methods

**Sample synthesis**. The high purity powers of Cu (99.95%) and Se (99.99%) were mixed homogeneously in an agate mortar to obtain a nominal composition of samples to be $Cu_2Se$. From the mixed powders with defined composition, a pellet with a diameter of 10 mm by the uniaxial cold pressing under 60 MPa was prepared. Subsequently, self-propagating high-temperature synthesis (SHS) method was carried out in a vacuum chamber at a base pressure of $\sim5\times10^{-3}$ Pa to obtain polycrystalline ingots solely containing $Cu_{2-\delta}Se$ α-phase[28–31]. The ingots were ground into powder again using the agate mortar and pestle to attain homogeneous powder of $Cu_2Se$. The obtained powder was sintered into dense ingots free from macroscopic voids by means of Spark Plasma Sintering (SPS) using a carbon mold and punches under a uniaxial pressure of 70 MPa during 3 min at 700 °C.

**Structure and chemical characterization**. The sintered samples were characterized by powder X-ray diffraction, using synchrotron radiation ($\lambda = 0.59986$ Å) and Debye−Scherrer geometry at discrete temperatures from room temperature to 477 K at Aichi Synchrotron Radiation Center (Japan), Beamline BL5S2. Chemical analyses using Electron probe micro-analyzer (EPMA) and Scanning Electron Microscope-Energy Dispersive X-ray spectroscopy (SEM-EDX) were carried out at room temperature by JEOL JXL-8230 and HITACHI SU 6600, respectively. In both cases, the energy of incident electrons used for the chemical analyses was 20 keV. More detailed information about chemical analyses is included in Supplementary Figures 11−13, Supplementary Table 1, and Supplementary Note 7.

**Thermoelectric property measurements**. The Seebeck coefficient $S(T)$ and electrical resistivity, $\rho(T)$, were measured over the temperature range from 300 to 500 K using our newly developed experimental setup, which is rather special and, therefore, explained in detail in Supplementary Figure 1, and Supplementary Note 1. $\rho(T)$ was measured by a standard four-probe method. To prevent the effects of thermal electromotive force and ionic conduction on the electrical resistivity, the AC current (0.05 Hz square wave) was used for the measurements. We also employed another system for measuring electrical resistivity using a tube furnace vacuumed to less than $10^{-2}$ Pa in which temperature was homogeneous.

Thermal conductivity was determined from thermal diffusivity, $D(T)$, and specific heat, $C_p(T)$, both of which were measured using Laser flash analysis (NETZSCH LFA 457). Almost the same results of $C_p(T)$ within the error of less than 10% was confirmed by differential scanning calorimeter (RIGAKU ThermoPlus EVO2 DSC8231). The sample density was determined by conventional Archimedes' principle using ethanol as working liquid.

## Data availability

The datasets obtained and/or analyzed in this study are available from the corresponding author on reasonable request.

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

## Acknowledgements

This work is partly supported as an advanced research program for energy and environmental technologies commissioned by the New Energy and Industrial Technology Development Organization (NEDO).

## Author contributions

D.B. is the main author who found the new behavior of Seebeck coefficient of $Cu_2Se$. R.S. and M.M. strongly contributed to the interpretation of the mechanism leading to the unusual behavior of Seebeck coefficient. K.D.-C. contributed to the theoretical calculation of electronic structure and Seebeck coefficient. S.C. and K.H. contributed to the preparation of samples and the measurements of thermoelectric properties. M.A., M.K., T.M. and Y.Y. contributed to the construction of scenario to interpret the unusual behavior of Seebeck coefficient as the main members of NEDO project. T.T. led the project, constructed the measurement system for Seebeck coefficient, and interpreted the observed phenomena. All authors discussed the results and commented on the manuscript.

## Additional information

**Competing interests:** The authors declare no competing interests.

