## [Peer Review File · Nature Communications]

Reviewers' comments:

Reviewer #1 (Remarks to the Author):

Colossal magnitude of Seebeck coefficients such as 10~20 mVK⁻¹ are often observed at the p-n junctions [Chapter 4, in "Thermoelectric Handbook, Macro to Nano, D.M. Rowe (editor)", CRC Press, 6000 Broken Sound Parkway NW, Suite 300, Boca Raton, FL, 33487, 912 pages, Taylor and Francis Group, 2006. ISBN: 0-8493-2264-2.]. Such large magnitudes are due to low carrier concentrations at the junctions, and accompanied by low electrical conductivities, resulting in low power factor and low dimensionless figure of merit.

Large Seebeck coefficients in p-n-p type conduction switching materials are also well-known, and also due to low carrier concentrations [Phys.Chem.Chem.Phys., 2015, 17, 10316].

The change in the majority carrier concentration is so dramatic in the depletion layer, and so are both the Seebeck coefficient and the electrical conductivity. Therefore it is very important to measure very carefully the Seebeck coefficient and the electrical conductivity at the same place and the same carrier concentration.

The authors claim in the manuscript that they observed Colossal Seebeck Effect in Metallic Cu_{1.97}Se. From the large Seebeck coefficients and the large electrical conductivities, they calculated very high power factor such as 2.3 Wm⁻¹K⁻². The materials with such high power factors should be very useful in many applications.

The measurement condition in the manuscript, however, is not constant over measurement time, and the same carrier concentration for the Seebeck and the electrical conductivity measurement is not guaranteed.

Authors should design better way to show both Seebeck coefficients and electrical conductivities are large at the same carrier concentration.

One way may be to measure Seebeck coefficient at the temperature having large Seebeck coefficient with larger temperature difference such as 40K~50K, not small such as 1K. With such a large temperature difference, the sample will be composed of parts with negative Seebeck and ones with positive, but still have colossal Seebeck coefficient based on the data in the manuscript. And authors will take ease to measure electrical conductivity at the same (or similar) carrier concentrations.

In addition, if authors measure both properties for elongated time period, and still observe high power factor over time, the results should be very useful.

Reviewer #2 (Remarks to the Author):

The authors report interesting behavior of colossal Seebeck effect (CSE) in Cu_{1.97}Se compound. However, it needs more experimental convincing evidences in order to prove that it is not an experimental artifact. In addition, the manuscript organization and quality of preparation should be significantly improved. Because the presentation quality of the manuscript is low and there is a misleading theoretical interpretation, I cannot recommend the publication of the manuscript as in this form but it can be considered to be published elsewhere when it is revised significantly and contains more improved experiment as follows:

1. The author should present the reproducibility of the measurement with repeated heat cycling (heating & cooling for several times).
2. If the CSE come from the vertical temperature gradient, it needs thickness-dependent experiment: For different samples with different thickness, please measure the vertical

temperature gradient and analyze the CSE with respect to thickness of the samples.

3. The CSE for temperature relaxation experiment might be interesting: Please measure the time-dependent Seebeck coefficient $S(t)$ for a step-functional heat pulse.

4. The comparison between theoretical calculation of Seebeck coefficient (Fig. 4B) w.r.t. chemical potential and temperature-dependent Seebeck coefficient (Fig. 2B) is totally wrong!! (Second paragraph of page 5: "Notably, despite that the... the calculated Seebeck coefficient $S(\mu)$ as a function of chemical potential show fairly good consistency with the experimental outcome $S(T)$...")

The positive and negative peaks of $S(T)$ within a temperature range $350 \text{ K} < T < 380 \text{ K}$ ($dT = 30 \text{ K}$) corresponds to very small chemical potential change ($d\mu = 0.003 \text{ eV}$). The chemical potential tuning by temperature is not likely because the energy scale of room temperature is almost close to the Fermi energy ($300 \text{ K} \sim 0.03 \text{ eV}$).

5. The author should clarify the qualitative or quantitative origin of the non-linear behavior of thermoelectric voltage with respect to temperature gradient (Fig. 2C).

For minor comments, I recommend a revision as follows:

1. (abstract, 2nd line) sing -> sign

2. (page 2, 2nd paragraph) Up to now, Bi₂Te₃ based materials... with a ZT of about 2.0 : The high ZT of 2.0 in Bi₂Te₃ based materials is controversial. So it needs the remark on the controversy. Please add the other reference as "Deng et al., Sci. Adv. 2018;4: eaar5606"

3. As the author stated in the manuscript, the thermoelectric properties of the Cu₂-dSe compounds are very sensitive with the Cu deficiency concentration. Please depict the final composition of the compounds rather than the nominal composition in the manuscript.

Reviewer #3 (Remarks to the Author):

This article reports that unusual thermoelectric transport properties of Cu_{1.97}Se polycrystalline bulk with mixed phases. It is interesting that the phase evolution and related thermoelectric properties is strongly correlated. Moreover, it is very surprising that colossal Seebeck coefficient and ZT are observed at around 350 K. The results are interesting and well organized, however, following issues should be clarified with additional observations prior to the further consideration for publication.

1. The setup for the Seebeck coefficient measurement, which is designed by the authors, seems to be available. However, it should be generalized thus the real photo for the measurement set-up with loaded sample is required. And the data for the accuracy should be provided. This might be carried out via the comparison of measured temperature dependence of Seebeck coefficient with measured data by a commercial system such as ZEM-3 by using standard sample (e.g. constantan, Bi-Te, etc.)

2. Considering the wide temperature range for the colossal Seebeck coefficient, the variation of Seebeck coefficient with temperature can be confirmed via the measurement by a commercial system (ZEM-3). The cyclic or static measurement around the phase transition temperature might be a possible approach.

3. Along with comment #1 and #2, it is very important to demonstrate the reliability of the measured Seebeck coefficient. The authors should provide the data to ensure the reliability via the repeated measurement for a sample.

4. Reproducibility of the sample is another critical issue. The authors should carried out the Seebeck coefficient measurement for many samples, and provide the results to secure the reliability of the data and reproducibility of the sample.

5. The authors provide the experimental and theoretical considerations for the occurrence of colossal Seebeck coefficient which is related with the phase transition. Claimed results and discussions are reasonable. Base on the insight of this study, can the authors surmise the thermoelectric transport properties of $\text{Cu}_{1.97+x}\text{Se}$ and $\text{Cu}_{1.97-x}\text{Se}$?

6. Detailed information for the microstructure of powders and bulks is missing.

7. In the description of the calculation of electronic structure of $\text{Cu}_{1.97}\text{Se}$, if any difference or similarity related to reported analysis, the proper references with related note should be included for a general readership.

8. What makes the difference in the temperatures for the peak values of electrical resistivity, heat capacity, and thermal conductivity?

First of all, I would like to greatly thank to all the referees for careful and deep reading of our previous manuscript, providing us with useful and beneficial comments. In this letter following, I provide responses to the comments and questions asked.

Reviewer #1,

[1st comment]

Colossal magnitude of Seebeck coefficients such as 10~20 mVK⁻¹ are often observed at the p-n junctions [Chapter 4, in “Thermoelectric Handbook, Macro to Nano, D.M. Rowe (editor)”, CRC Press, 6000 Broken Sound Parkway NW, Suite 300, Boca Raton, FL, 33487, 912 pages, Taylor and Francis Group, 2006. ISBN: 0-8493-2264-2.]. Such large magnitudes are due to low carrier concentrations at the junctions, and accompanied by low electrical conductivities, resulting in low power factor and low dimensionless figure of merit.

[Our response]

The large magnitude of Seebeck coefficient is not very interesting by itself, because Seebeck coefficient is a function of carrier concentration and becomes larger with decreasing carrier concentration. As the first referee suggested, we can observe a large magnitude exceeding 1mVK⁻¹ even in the semiconducting Si. The most important point of our finding is a large magnitude of Seebeck coefficient simultaneously observed together with small values of electrical resistivity, and thus, the finding is surely of great importance .

[2nd comment]

Large Seebeck coefficients in p-n-p type conduction switching materials are also well-known, and also due to low carrier concentrations [Phys.Chem.Chem.Phys., 2015, 17, 10316]. The change in the majority carrier concentration is so dramatic in the depletion layer, and so are both the Seebeck coefficient and the electrical conductivity. Therefore, it is very important to measure very carefully the Seebeck coefficient and the electrical conductivity at the same place and the same carrier concentration.

[Our response]

Honestly speaking, we had not been aware of p-n-p transition observed for some materials, and greatly appreciate the comments. We mentioned the similarity and difference between the previously reported p-n-p transitions in the manuscript. The main difference is the electrical resistivity at the temperature range of a large Seebeck coefficient. The previously reported materials possessing p-n-p transition were highly resistive over the whole temperature range in sharp contrast to the metallic electrical conduction in our

samples. We also emphasized in the text that we do not have an intermediate phase, that is responsible for the large magnitude of negative Seebeck coefficient, between the low-temperature and high-temperature phase in the previously reported materials.

As for the evaluation of electrical resistivity and the Seebeck coefficient evaluation, during our measurements, both quantities were always acquired simultaneously by the same electrodes as our designed Seebeck setup has this ability. Therefore, the measured values of resistivity and Seebeck coefficient attained at particular temperature reflect the same carrier concentration of the measured Cu_2Se system at particular place, set temperature and in our temperature distribution case.

[3rd comment]

The authors claim in the manuscript that they observed Colossal Seebeck Effect in Metallic $\text{Cu}_{1.97}\text{Se}$. From the large Seebeck coefficients and the large electrical conductivities, they calculated very high power factor such as $2.3 \text{ Wm}^{-1}\text{K}^{-2}$. The materials with such high power factors should be very useful in many applications. The measurement condition in the manuscript, however, is not constant over measurement time, and the same carrier concentration for the Seebeck and the electrical conductivity measurement is not guaranteed. Authors should design a better way to show both Seebeck coefficients and electrical conductivities are large at the same carrier concentration. One way may be to measure Seebeck coefficient at the temperature having large Seebeck coefficient with larger temperature difference such as $40\text{K}\sim 50\text{K}$, not small such as 1K . With such a large temperature difference, the sample will be composed of parts with negative Seebeck and ones with positive, but still, have colossal Seebeck coefficient based on the data in the manuscript. And authors will take ease to measure electrical conductivity at the same (or similar) carrier concentrations.

[Our response]

We measured electrical resistivity and Seebeck coefficient on the same sample in the same experimental apparatus simultaneously. This means that the carrier concentration of the sample is always the same for both electron transport properties. The idea of larger temperature difference is not acceptable because the thermoelectric voltage is obtained by the integration over the temperature range. The wider temperature difference greatly broadens the significant temperature dependence of the Seebeck coefficient, despite that it is the main point of arguments. We explained the theoretical description and the scientific idea via equations for the Seebeck coefficient in Supplementary information section I. in detail, and I hope that the referee understands its essence by reading it.

[4th comment]

In addition, if authors measure both properties for elongated time period, and still observe high power factor over time, the results should be very useful.

[Our response]

It is of great importance to prove whether the colossal value of power factor is sustainable for a longer period of time. Unfortunately, however, the current experimental setup does not allow us to continuously measure the phenomena with keeping the temperature. Instead of that, we measured the properties for several times on heating after cooling and confirmed that it is reproducible. The results are included in the section III. of Supplementary information.

Reviewer #2,

The authors report interesting behavior of colossal Seebeck effect (CSE) in Cu_{1.97}Se compound. However, it needs more experimental convincing evidences in order to prove that it is not an experimental artifact. In addition, the manuscript organization and quality of preparation should be significantly improved. Because the presentation quality of the manuscript is low and there is a misleading theoretical interpretation, I cannot recommend the publication of the manuscript as in this form but it can be considered to be published elsewhere when it is revised significantly and contains more improved experiment as follows:

[1st comment]

The author should present the reproducibility of the measurement with repeated heat cycling (heating & cooling for several times).

[Our response]

It has been done and included in the Supplementary information. Additionally, we show a set of data of Seebeck coefficient data in the main text indicating the same abnormal behavior and colossal values.

[2nd comment]

If the CSE come from the vertical temperature gradient, it needs thickness-dependent experiment: For

different samples with different thickness, please measure the vertical temperature gradient and analyze the CSE with respect to thickness of the samples.

[Our response]

We had already realized this point and planned to write it in the second paper. However, as suggested, we decided to include data it in this manuscript in its Supplementary information. We definitely observe strong thickness dependence that proves our interpretation of the unusual behavior of Seebeck coefficient in Cu_2Se .

[3rd comment]

The CSE for temperature relaxation experiment might be interesting: Please measure the time-dependent Seebeck coefficient $S(t)$ for a step-functional heat pulse.

[Our response]

As mentioned in the main text as well as in Supplementary information, the measurements using our Seebeck setup are based on step-function heating using heat pulses, where the voltages defining temperatures $T_1(t)$, $T_2(t)$ and $T_3(t)$, as labeled in the supplementary information section I., are captured as functions of time, however, it is very difficult to keep steady temperature at position of negative peak as it is very narrow temperature range (\sim a few of Kelvin). The region of the abnormal behavior is captured in our data continuously on heating, thus setting the temperature at exactly the peak temperature position and measure the Seebeck coefficient as a function of time to confirm time stability and reliability is really hard, even almost impossible, task. Besides, the peak position varies within 1 - 2 Kelvin from sample to sample (considering error and accuracy) which indeed makes situation even harder to comply.

Therefore for now, we keep this point unsolved in this particular stage, but will try to achieve the measurement in near future.

[4th comment]

The comparison between the theoretical calculation of Seebeck coefficient (Fig. 4B) w.r.t. chemical potential and temperature-dependent Seebeck coefficient (Fig. 2B) is totally wrong!! (Second paragraph of page 5: “Notably, despite that the... the calculated Seebeck coefficient $S(\mu)$ as a function of chemical potential show fairly good consistency with the experimental outcome $S(T)$...”)

The positive and negative peaks of $S(T)$ within a temperature range $350 \text{ K} < T < 380 \text{ K}$ ($dT = 30 \text{ K}$) corresponds to very small chemical potential change ($d\mu = 0.003 \text{ eV}$). The chemical potential tuning by

temperature is not likely because the energy scale of room temperature is almost close to the Fermi energy (300 K \sim 0.03 eV).

[Our response]

We know the temperature dependence of chemical potential is small for a given electronic structure following the above-explained arguments. However, the referee's arguments (#4) are valid only for one particularly given electronic structure in a single phase, and a scenario with two separated phases coexisting together is not justifiable anymore. The situation has to be more complicated as usually and naively followed, in other words, the trivial energy-to-temperature equivalency cannot be used at all. The point is that we have two phases in one sample due to the unusual temperature distribution, and the chemical potentials of electrons and Cu ions in the two phases are different to motivate electrons and/or Cu ions moving across the phase boundaries. In such a case, chemical potential moves drastically with varying temperature with variation of the volume fraction of two phases. This is nothing but a speculation, however, qualitatively account for the phenomena, as already discussed in the main text in detail.

[5th comment]

The author should clarify the qualitative or quantitative origin of the non-linear behavior of thermoelectric voltage with respect to the temperature gradient (Fig. 2C).

[Our response]

The non-linear behavior of thermoelectric voltage is caused by the significant temperature dependence of the Seebeck coefficient. By reading the theoretical description addressed via the equations used in our analyses of measurements written in Supplementary information section I., the referee would understand easily the physics responsible for the non-linear behavior of thermoelectric voltage. If we assume for simplicity the Seebeck coefficient linearly varying with temperature in the temperature gradient applied through a sample, the thermoelectric voltage becomes $V = \int_t^{t+dt} \alpha T dT = 0.5\alpha\{(t + dt)^2 - t^2\} = 0.5\alpha\{(dt)^2 + 2tdt\}$. This tendency clearly provides a non-linear temperature dependence of thermoelectric voltage.

[6th comment]

For minor comments, I recommend a revision as follows:

1. (abstract, 2nd line) sing -> sign

2. (page 2, 2nd paragraph) Up to now, Bi₂Te₃ based materials... with a ZT of about 2.0 : The high zT of 2.0 in Bi₂Te₃ based materials is controversial. So it needs the remark on the controversy. Please add the other reference as “Deng et al., Sci. Adv. 2018;4: eaar5606”

3. As the author stated in the manuscript, the thermoelectric properties of the Cu₂-dSe compounds are very sensitive with the Cu deficiency concentration. Please depict the final composition of the compounds rather than the nominal composition in the manuscript.

[Our response]

According to the comments, we revised the manuscript appropriately. See the manuscript with colored markers.

Reviewer #3,

This article reports that unusual thermoelectric transport properties of Cu_{1.97}Se polycrystalline bulk with mixed phases. It is interesting that the phase evolution and related thermoelectric properties are strongly correlated. Moreover, it is very surprising that colossal Seebeck coefficient and ZT are observed at around 350 K. The results are interesting and well organized, however, following issues should be clarified with additional observations prior to the further consideration for publication.

[1st comment]

The setup for the Seebeck coefficient measurement, which is designed by the authors, seems to be available. However, it should be generalized thus the real photo for the measurement set-up with loaded sample is required. And the data for the accuracy should be provided. This might be carried out via the comparison of measured temperature dependence of Seebeck coefficient with measured data by a commercial system such as ZEM-3 by using standard sample (e.g. constantan, Bi-Te, etc.)

[Our response]

We added the photo of the measurement system in Fig.S1 of Supplementary information section I. The accuracy of measurements was confirmed by the other samples, constantan, and higher manganese silicide, and the results were already shown in Figs. S8 and S9 in supplementary information section VI.

[2nd comment]

Considering the wide temperature range for the colossal Seebeck coefficient, the variation of Seebeck coefficient with temperature can be confirmed via the measurement by a commercial system (ZEM-3). The cyclic or static measurement around the phase transition temperature might be a possible approach.

[Our response]

We also measured the Seebeck coefficient of our samples with Thermal Transport Option of Physical Properties Measurement System purchased by Quantum Design, Inc. The results were shown in Fig.S7 in Supplementary information section V., showing a good consistency with the reference data but not unveiling any of its abnormal behavior reported in this study. This fact indicates that the unusual temperature gradient plays an essential role in realizing the huge magnitude of Seebeck coefficient of Cu_2Se .

[3rd comment]

Along with comment #1 and #2, it is very important to demonstrate the reliability of the measured Seebeck coefficient. The authors should provide the data to ensure the reliability via the repeated measurement for a sample.

[Our response]

We add new data of electrical resistivity and Seebeck coefficient measured on one particular sample with a defined thickness on cycles as shown in Figs. S4 and S5 of Supplementary information section III. The same behavior was observed in the multiple measurements.

[4th comment]

Reproducibility of the sample is another critical issue. The authors should carry out the Seebeck coefficient measurement for many samples, and provide the results to secure the reliability of the data and reproducibility of the sample.

[Our response]

We measured the behavior more than a hundred times, and still observed the abnormal behavior of Seebeck coefficient reported in this manuscript. To show the reproducibility of the presented phenomenon, we added representative and typical data of Seebeck coefficient obtained on Cu_2Se samples as shown in Fig. 3 of the main text, and S5, and S6 of Supplementary information section III. and IV.

[5th comment]

The authors provide the experimental and theoretical considerations for the occurrence of colossal Seebeck coefficient which is related with the phase transition. Claimed results and discussions are reasonable. Base on the insight of this study, can the authors summarize the thermoelectric transport properties of $\text{Cu}_{1.97+x}\text{Se}$ and $\text{Cu}_{1.97-x}\text{Se}$?

[Our response]

A lot of work has been already done on the Cu_{2+x}Se and Cu_{2-x}Se systems in previous studies as it was under an intense focus in the Thermoelectric community, however, not showing the phenomenon argued in this study. Therefore, to summarize previous results does not provide any connection between our and previously reported results being not interrelated.

As for our own data, we have also considered that the composition dependence of colossal Seebeck effect is very important. Despite that we have already started, the full investigation has not accomplished yet. Therefore it will be reported in the next paper.

[6th comment]

Detailed information for the microstructure of powders and bulks is missing.

[Our response]

The detailed information about the crystal structure of our samples has been already written in Supplementary information, section VIII. We performed SEM-EDX XRD, EPMA line analyses, and Synchrotron Rietveld analyses. We indeed confirmed that crystal structure properties are same for both power as studied by the Synchrotron radiation, and pellet invariants checked in our laboratory by XRD (Bruker D8 Advanced, $\lambda=1.5418 \text{ \AA}$) of Cu_2Se . All these experiments and results have been shown in the main text and/or Supplementary information.

[7th comment]

In the description of the calculation of electronic structure of $\text{Cu}_{1.97}\text{Se}$, if any difference or similarity related to reported analysis, the proper references with related note should be included for a general

readership.

[Our response]

We did not find reports on the electronic structure of Cu_2Se . That is presumably because the low-temperature phase has split sites and the high-temperature phase has very low occupancies in some sites in relation with the ionic conduction. These characteristics definitely make the reliability of band calculation worse, and hence, only little attempts were done to calculate its electronic structure.

[8th comment]

What makes the difference in the temperatures for the peak values of electrical resistivity, heat capacity, and thermal conductivity?

[Our response]

The peak temperature value of heat capacity, thermal conductivity and resistivity (measured in tube furnace) are almost identical within our accuracy of the measurements. As for the peak temperature in the case of measurement carried out by the Seebeck setup, the reason for the different temperature of peaks was written in the main text as follows. In our measurement system, the temperature was measured at the top surface, but the electrical current passes through the bottom of the sample where the temperature is definitely higher than the top surface. This configuration makes the measured peak of electrical resistivity artificially lower.

I strongly hope that the referees understand our arguments and that the manuscript is accepted for publication in Nature Communications.

Yours sincerely

Tsunehiro Takeuchi

竹内 恒博

Professor

Toyota Technological Institute,

October 4th, 2018

Reviewers' comments:

Reviewer #1 (Remarks to the Author):

In the authors' responses, "The most important point of our finding is a large magnitude of Seebeck coefficient simultaneously observed together with small values of electrical resistivity, and thus, the finding is surely of great Importance."

High power factor of large magnitudes of Seebeck coefficient and small values of electrical resistivity is of great importance. In the temperature range having peak power factor, the Seebeck coefficient changes sharply with temperatures, which is typically accompanied by sharp change in the carrier concentration, and the electrical resistivity. Authors measured Seebeck coefficients with a very small step size of temperatures showing sharp evolution of Seebeck coefficients. However, they didn't show data of electrical resistivity at the same small step of temperatures. If they missed observing abrupt change in electrical resistivity, and there were a sharp increase in the electrical resistivity, the power factor were not so high, and the observation would be simply one of well-known phenomena. Therefore, authors should check electrical resistivity very carefully with very small step of temperatures, to show no sharp increase in electrical resistivity in the temperature range.

Reviewer #2 (Remarks to the Author):

The authors clarified the issues what I raised in the previous referee's comment. So, I recommend this manuscript is suitable to be published as in this form.

Reviewer #3 (Remarks to the Author):

This article reports that unusual thermoelectric transport properties of $\text{Cu}_{1.97}\text{Se}$ polycrystalline bulk. It is very surprising that unexpected high ZT are observed at around 350 K. The authors addressed well to the issues which I raised and provide the experimental data to ensure the reliability of measurement. Thus I recommend publication of this paper without further revision.

I would like to greatly appreciate all the reviewers to carefully read our revised manuscript and especially of the first reviewer for the useful comment about electrical resistivity. I shall write in this letter the responses to the comments.

Reviewer #1,

[Comment]

Authors measured Seebeck coefficients with a very small step size of temperatures showing sharp evolution of Seebeck coefficients. However, they didn't show data of electrical resistivity at the same small step of temperatures. If they missed observing abrupt change in electrical resistivity, and there were a sharp increase in the electrical resistivity, the power factor were not so high, and the observation would be simply one of well-known phenomena. Therefore, authors should check electrical resistivity very carefully with very small step of temperatures, to show no sharp increase in electrical resistivity in the temperature range.

[Our response]

We totally agree with the first referee in the point that the variation of carrier concentration in the low temperature phase at the top surface of sample might cause the sharp peak in the electrical resistivity at the condition of the largest Seebeck coefficient. Our previous data certainly lack the detailed information about the electrical resistivity with poor temperature resolution. Therefore, we performed additional experiment to confirm whether any unusual increase of electrical resistivity is observable or not.

Since the simultaneous measurements of electrical resistivity and Seebeck coefficient prevented us from measuring electrical resistivity with fine temperature step, therefore we measured the electrical resistivity without measuring Seebeck coefficient in the same experimental setup with the same temperature gradient of colossal value of Seebeck coefficient. As a result, we did not observe any unusual increase in the electrical resistivity and therefore we did not make any revision on the main text, but add one figure (**Figure S6**) in which the result of newly performed experiments are displayed.

The obtained result means that the low temperature phase of nearly insulating condition is thin enough to make the increase of electrical resistivity less emphasized. I strongly hope that the first referee and the editors understand our arguments on new data, and that the manuscript is now acceptable for publication in Nature Communications.

Yours sincerely

Tsunehiro Takeuchi

竹内 恒博

Professor

Toyota Technological Institute,

November 8th, 2018

REVIEWERS' COMMENTS:

Reviewer #1 (Remarks to the Author):

I appreciate author's efforts to get detailed electrical resistivity measurements. Now I recommend this manuscript for publication in Nature Communications.

I found one typo error in the Supplementary Materials.

In the page 8,
Since our experimental setus of simultaneous -> setup

Thank you